# Sensitivity and Specificity of two rapid tests for the diagnosis of infection by *Trypanosoma cruzi* in a Colombian population

**Sandra Helena Suescún-Carrero**[1]*, **Lyda Pilar Salamanca-Cardozo**[2], **María-Jesus Pinazo**[3], **Lluis Armadans-Gil**[4]

1 Grupo de Investigación del Laboratorio de Salud Pública de Boyacá, Secretaria de Salud de Boyacá, Tunja, Colombia, Doctorado en Metodología de la Investigación Biomédica y Salud Pública, Universitat Autònoma de Barcelona, Barcelona, Spain, 2 Grupo de Investigación del Laboratorio de Salud Pública de Boyacá, Secretaria de Salud de Boyacá, Tunja, Colombia, 3 Barcelona Institute for Global Health (ISGlobal), Hospital Clinic—University of Barcelona, Barcelona, Spain, 4 Servicio de Medicina Preventiva y Epidemiología, Hospital Universitari Vall d'Hebron—Universitat Autònoma de Barcelona, Barcelona, Spain

* sandrahsc@yahoo.com

## Abstract

### Objective

To evaluate diagnostic precision of two rapid diagnostic tests (RDT's) on patients with chronic Chagas disease.

### Methodology

Prospective study with the following inclusion criteria: subjects older than 3 years, signed informed consent. Exclusion criterion: subjects could not have previously received treatment for infection with *T. cruzi*. The study population were participants in a screening process undertaken in rural and urban zones of the department Boyacá, Colombia. Two RDT's were performed to all participants: the Chagas Detect Plus InBios (CDP) and the Chagas Stat-Pak (CSP) and as a reference standard the ELISA Chagas III GrupoBios and the Chagas ELISA IgG+IgM I Vircell tests were used. In the case of discordant results between the two ELISA tests, an indirect immunofluorescence was done.

### Results

Three hundred-five (305) subjects were included in the study (38 patients with leishmaniasis), of which 215 tested negative for *T cruzi* and 90 tested positive according to the reference standard. The sensitivity of the RDT's were 100% (CI 95% 95.9–100), and the specificity of the CDP was 99.1% (CI 95% 96.6–99.8) and for CSP was 100% (CI 95% 98.3–100). The agreement of CDP was 99.5% and for CSP was 100% with Kappa values of (k = 99.1; CI 95% 92.6–99.8%) and (k = 100; CI 95% 94.3–100), respectively. RDT's did not present cross-reactions with samples from patients who were positive for leishmaniasis.

**Data Availability Statement:** All relevant data are within the manuscript and its Supporting Information files.

**Funding:** This project was financed by the Government of Boyacá - Secretary of Health of Boyacà through the Departmental Laboratory of Public Health (BP and P Registry 2019004150406) https://www.boyaca.gov.co/. SSC and LPSC had a contract from the Government of Boyacá in the departments of research (1759-2019 / 623-2020 / 2023-2020) and diseases transmitted by vectors of the LDSP (1750-2019 / 1359-2020 / 1969-2020). Research by LAG was funded by the Preventive Medicine and Epidemiology Service, Hospital Universitari Vall d'Hebron - Autonomous University of Barcelona, Barcelona, Spain. The funders had no role in study design, data collection and analysis, decision to publish, or preparation of the manuscript.

**Competing interests:** No.The authors have declared that no competing interests exist.

## Conclusions

The findings demonstrate excellent results from the RDT's in terms of validity, safety, and reproducibility. The results obtained provide evidence for the recommendation for using these tests in a Colombian epidemiological context principally in endemic areas in which laboratory installations necessary to perform conventional tests are not available, or they are scarce and to help in diagnosing chronic Chagas disease in order to provide access to treatment as soon as possible.

## Author's summary

Chagas is a disease caused by the parasite *Trypanosoma cruzi*, and is one of the most important public health concerns affecting the population of Latin America. This disease presents an acute phase that generally goes undiagnosed and a chronic phase with cardiac manifests principally, is diagnosed through serological tests that are not available in the majority of regions endemic for Chagas disease (CD), the results may take weeks to be returned due to logistical and operational reasons that comprise the main obstacles in initiating treatment of the disease. In the present article, quality indices of two RDT's were evaluated during a field study in the department of Boyacá Colombia, these tests are easy to administer, require only minimal quantities of sample, provide rapid results and do not require electrical equipment or refrigeration. The sensitivity of the two RDT's was 100% and the specificity of CDP was 99.1% and for CSP was 100% according to reference tests. The results obtained support the recommendation of using RDT's in order to help diagnose chronic Chagas disease and thus to improve access to treatment for the populations studied as soon as possible.

## Introduction

Tripanosomiasis americana, or CD, is caused by the parasite *Trypanosoma cruzi (T. cruzi)*, an endemic microorganism in Latin America and transmitted by vectors from several species of the subfamily Triatominae [1]. *T. cruzi* may be transmitted in various ways, but is mainly transmitted through skin and/or mucus membrane contact with the feces of infected triatomines, which, when they bite, deposit fecal matter on the host containing metacyclic trypomastigotes that may enter through the orifice of the bite by scratching, through discontinuity in the skin, or through the ocular and/or nasal conjunctiva. Other methods of transmission include: oral transmission, which occurs through the consumption of foods or beverages in most cases contaminated with the feces of infected triatomines; transfusional transmission, which occurs from the presence of live tripomastigotes and infectants in the blood of donors; accidental transmission through puncture or other types of contact with material contaminated with the parasite; congenital transmission, in which the parasite is transmitted through the placenta to the fetus [2] and transmission from organ transplants due to the fact that, as with transfusions, transplants with organs from donors who live in endemic areas may contain the parasites and disseminate parasitosis to an immunosuppressed host [3].

There are two phases in the natural history of CD: acute and chronic. In the acute phase, individuals who do not receive specific treatment evolve to the chronic phase of the infection. In this phase, between 50 and 70% of infected persons do not develop any harmful corporal

effects, and may remain in this state for the rest of their lives (indeterminate form of the infection). However, after 20–30 years or more, 30–50% of these individuals present primarily cardiac and digestive alterations that may cause significant morbidity and even death in some cases [3]. Due to the fact that clinically, the acute infection is usually asymptomatic or pauci-symptomatic, the infection is diagnosed in this phase only 1 to 2% of the time, which is unfortunate, as the efficacy of treatment at this stage is much better [4]. Thus, the majority of cases are diagnosed in the chronic phase, most often using serological techniques [3].

From an epidemiological viewpoint on the regional level, CD is one of the principal public health problems that affects the population of Latin America. According to the World Health Organization (WHO), there are between 6 and 7 million people infected with the parasite in the world from endemic zones of 21 Latin American countries, and 52 million live in zones of vector transmission risk [5]. CD is the third most common parasitic infection in the world, and is an emerging disease in Spain, the United States, and other countries where it is not endemic [6]. In Colombia, its prevalence is reported to be between 700,000 and 1,200,000 inhabitants infected and 8,000,000 individuals at risk of acquiring the infection [7,8,9]. The departments with the greatest degrees of endemism are Santander, Norte de Santander, Cundinamarca, Boyacá, Casanare and Arauca; and most recently, communities in the Sierra Nevada de Santa Marta [8,9,10]. Infection by *T. cruzi* has also been detected along the entire length of the Magdalena river, in the Catatumbo region, in the piedmont of the Llanos Orientales in the Serranía de la Macarena [9,10]. In studies done in different departments of Colombia, the following prevalences have been reported: Sierra Nevada de Santa Marta 36.9% [10]; Casanare 16.9% [11], Boyacá 7.8% [12], Santander 3.2% [13], Guaviare 2.07, Vaupés 0.79% and Amazonas with 0.09% [14].

In the Americas, up to 99% of cases of CD and more than 90% of cases of CD in Europe are undiagnosed [15,16]. Colombia faces several barriers to the diagnosis and treatment of *T. cruzi* infection, as only 1.2% of the at-risk population has been examined, while only between 0.3 and 0.4% have received etiological treatment [17]. Serological diagnosis in the chronic phase of CD is based on the detection of circulating antibodies through conventional reactions such as the enzyme-linked immunosorbent assay (ELISA) test, Indirect Immunofluorescence (IIF), Indirect Hemagglutination (IHA), Chemoluminescence (ChLIA) and Western blot/Immunoblot [18]. The WHO accepts as positive results two tests based on different immunological principles as laboratory diagnostic parameters and a third test in cases in which the first two tests were discordant [19,20]. Currently, the serological diagnostic algorithm in Colombia includes the use of two ELISA tests in series, with different principals and different types of antigens (total antigens, synthetic peptides or recombinants) with high sensitivity and specificity [18]. These laboratory tests require qualified personnel as well as equipment and infrastructure that are not available in the majority of zones in which the disease is endemic. Also, the results of tests may take weeks to be delivered due to logistical or operational constraints. The lack of access to diagnostics is thus one of the main obstacles to beginning treatment for CD [21].

In 2007, in several scientific forums and in 2010 in the 63rd World Health Assembly, the urgent need was expressed for new, simpler diagnostic tools, ideally through the use of Rapid Diagnostic Testing (RDT), for the detection of infection by *T. cruzi* / CD; in order to decrease underdiagnosis in remote areas in which diagnosis is not accessible using conventional techniques and there is a need for more timely treatment [22,23]. Several RDT's exist for the detection of infection by *T.cruzi* that are easy to use, only require minimal quantities of sample (whole blood), provide rapid results; do not require specialized laboratory personnel, electrical equipment and can be stored at room temperature, thus making them ideal for field studies [24]. Studies have been carried out in several countries evaluating rapid tests that have

obtained different results in terms of sensitivity and specificity according to the country, with factors such as geographic variation (which could be related to different strains of the parasite), incidence, rate of transmission and prevalence of the disease, playing important roles in the outcomes obtained [25,26].

It is thus important to understand the efficacy of these tests in terms of sensitivity and specificity in the Colombian epidemiological context in order to broaden diagnostic programs and to allow access to more rapid treatment in order to prevent patients from developing cardiac alterations. Due to the above, the objective of the present study is to evaluate the validity and safety of two RDT's: Chagas Detect Plus (InBios) CDP and Chagas Stat-Pak (Chembio) CSP, using total antigen ELISA tests and recombinant antigen ELISA tests as reference standard.

## Methods

### Ethics statement

The study was approved by the bioethics committee of the Universidad de Boyacá according to memorandum CB 039–2019 of May 24, 2019. Signed informed consent was obtained for all the participants, and signed consent was obtained by a legal representative for participants under 18 years old. All participants received their individualized test results and those who received positive results received medical counseling regarding the state of their infection and the Boyacá LDSP was notified. http://dx.doi.org/10.17504/protocols.io.bttpnnmn

### Study design, location, and participant sampling

A blind prospective observational study was developed, the inclusion criteria of which was that participants had to be older than 3 years of age and had to sign an informed consent form. The exclusion criterion of the study was to have previously received treatment for a *T. cruzi* infection. The study was implemented in urban and rural areas in seven municipalities of the department of Boyaca, of which, five: Soatá, Tipacoque, Chitaraque, Moniquirá and Miraflores are endemic zones for infection with *T. cruzi*, and two municipalities: Otanche and San Pablo de Borbur, which are endemic for Leishmaniasis, but not for *T. cruzi*. The department of Boyacá has a population of 1,278,000 people (664,560 men and 613,440 women) [27]. The calculation for sample size was done using the GRANMO program [28]: in order to estimate with a precision of 5%, a confidence level of 95%, a sensitivity and specificity of 90% [26], the necessary sample size was of 305 subjects.

The study population was composed of participants that attended field screenings done in the rural and urban zones of the seven municipalities included in the study. The field study was planned as to have a support team in each of the zones to take charge of information and education activities. Participants in the study were brought together by technicians from the vectors program of the Boyacá Secretary of Health from June to November 2019, and signed up to the program through their own initiative. The details of the study were explained to them, they signed an informed consent form, and a sample of whole blood was taken through puncture of a finger to be used in the field-administered RDT's. During the same visit, 5 mL of blood was extracted through venopuncture for the isolation of serum and was transported to the Departmental Health Laboratory of Boyacá (LDSP) so that the two ELISA tests could be completed and Indirect Immunofluorescence assay test (IIF) if necessary, according to the diagnostic algorithm for CD authorized in Colombia.

### Conventional serological tests and Rapid Diagnostic Tests (RDT's)

The reference diagnostic strategy utilized in the present study is based on conventional serological tests suggested by the National Parasitology Reference Laboratory of the National

Health Institute (Instituto Nacional de Salud, or INS) of Colombia [20], and is routinely used in the CD monitoring program of the LDSP. All samples were analyzed using the ELISA Chagas III GrupoBios tests [informed sensitivity, 100%; specificity, 100%] [29] and with the Chagas ELISA IgG+IgM I Vircell tests [informed sensitivity, 100%; specificity, 98%] [30,31]. In cases in which discordant results were found between the two ELISA tests, an IIF test was done [32], which uses epimastigotes of the Colombian *T. cruzi* strains DTU and TcI as antigens, when the results of this technique were positive, case was considered to be confirmed. The tests were done in serum samples in the LDSP, following the manufacturer's instructions; the laboratory complies with the internal quality control standards of the method, using material for quality control of the MQC and with external quality controls and aptitude tests done by the INS and Proasecal. The results of the tests were interpreted as either positive or negative.

There are several RDT's for detecting infection by *T.cruzi* which have the advantages described above over conventional tests and that have high sensitivities and specificities according to several authors [26,33,34]. Thus, in the present study, all samples were analyzed using two immunochromatographic diagnostic tests based on different antigens: Chagas Detect Plus (InBios International Inc., Seattle, USA) CDP [35], which uses a multi-epitope antigen, and Chagas Stat-Pak (Chembio Inc., Medford USA) CSP [36], which employs a combination of recombinant proteins. Small quantities of whole blood from finger puncture were used to carry out the RDT's. In order to evaluate cross-reactivity with leishmaniasis, participants from the municipalities of Otanche and San Pablo de Borbour were convened, they were diagnosed by the program of diseases transmitted by vectors of the LDSP with cutaneous leishmaniasis confirmed through the identification of the parasites from smears from cutaneous lesions using microscopy.

## Data analysis

For descriptive statistics of the categorical variables, frequencies were calculated; while for the continuous variables (age), the mean and standard deviation (SD) were calculated. In order to estimate the validity of the RDT's, the sensitivity, the specificity, the global value of the test, and the diagnostic efficiency (DE) (probability that the individual was classified correctly by the test) were used. To estimate the predictive values of the RDT's, a positive predictive value (PPV) (the proportion of patients who resulted positive who really had the disease) and a negative predictive value (NPV) (or the proportion of patients who resulted negative who did not have the disease) were found. The capacity of the RDT's to confirm or exclude CD was evaluated through a positive likelihood ratio (LR+) and a negative likelihood ratio (LR-).

In order to estimate the agreement between the two RDT's, a Kappa (k) coefficient that calculates the agreement between results, adjusted by the agreement expected by chance was used. The concordance values needed for a RDT to be considered reliable were: if K > 0.8; a value of k >0.8 and ≤0.9 indicated very high agreement, while 0.9> k ≤1.0 was considered to be excellent [37]. The level of statistical significance was established at 0.05. Data was analyzed using the R statistical program [38].

## Results

### Flow and demographic characteristics of participants

The present study was completed between June and November of 2019, the RDT's and the sample of blood through venopuncture for reference tests were done during the same visit. Three-hundred five (305) participants were included in the study according to the inclusion and exclusion criteria, of which 215 were negative for *T cruzi* and 90 were positive according

to the reference criteria. Two (2) discordant samples were obtained by the RDT's, and these were found to be negative by the reference tests (Fig 1).

Of the participants, 60.7% were women, 39.3% were men, the median age was 47.1 years old (SD = 18.5), the youngest participant was 3 years old and the oldest participant was 80

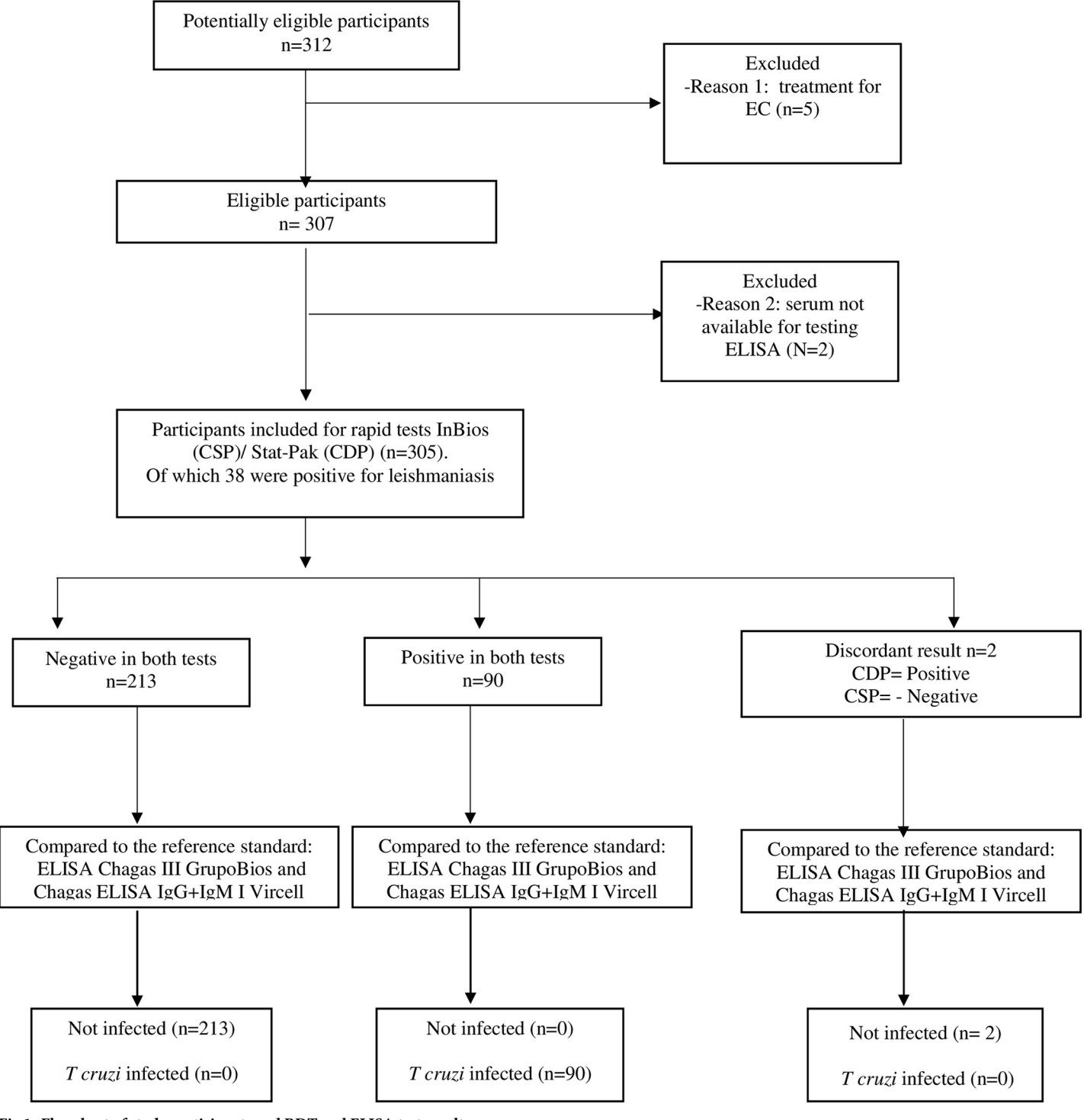

**Fig 1. Flowchart of study participants and RDT and ELISA test results.**

years old. The distribution by age range was: between 1 and 20 years old, 12.8%; 21 to 40 years 18.6%; 41 to 60 years 42.3% and from 61 to 80 years old 26.3%. The percentages of participants from each municipality were: Soatá 34.4%; Tipacoque 21.0%; Chitaraque 18.7%; Otanche 10.5%; Moniquira 9.8%; Miraflores 3.6% and San Pablo de Borbur 2.0%.

The seroprevalence obtained in the population of the seven municipalities of the Boyacá department was 29.5% (90/305); and according to sex, men presented a higher seroprevalence 31.6% than women 28.1%. The age category with the highest prevalence was from 61 to 80 years old, at 52.5%; followed by 41 to 60 year-olds with 34.9%. The municipalities with the highest percentages of infections by *T. cruzi* were Miraflores with 72.2%, followed by Moniquirá with 56.7% and Chitaraque with 38.6% (Table 1). The participants who were positive for CD were remitted to a CD observation program that is administered by the Secretary of Health of Boyacá for medical evaluation and management.

## Evaluation of conventional serological tests

There were two discordant tests within the ELISA tests conducted (participants who were not infected with *T. cruzi* who had Leishmaniasis) that were processed using a third IIF test according to the diagnostic algorithm for CD [18,19], obtaining titles with fluorescence that were very close to, but below the reference value (Table 2), so a new sample was taken from the two participants on which conventional tests and RDT's were performed after 30 days, as described in the guidelines of the Laboratory Oversight guide for *T cruzi* of the INS [20], these obtained positive results again for the ELISA total antigen test, negative results for the recombinant antigen ELISA test, and unreactive in the IIF test, and were subsequently considered to be negative for CD.

## Evaluation of RDT's

The CDP test detected two false positive cases that were confirmed to be negative with conventional serological tests, while the results of the CSP test agreed with the results obtained in the reference tests (Table 3).

**Table 1. Characterization of the population included in the study (n = 305).**

| Variables | Positive | | Negative | | Total |
|---|---|---|---|---|---|
| | n | (%) | n | (%) | |
| Sex | | | | | |
| Female | 52 | (28.1) | 133 | (71.9) | 185 |
| Male | 38 | (31.6) | 82 | (68.4) | 120 |
| Age Range | | | | | |
| 1–20 | 0 | (0) | 39 | (100) | 39 |
| 21–40 | 3 | (5.3) | 54 | (65.1) | 57 |
| 41–60 | 45 | (34.9) | 84 | (65.1) | 129 |
| 61–80 | 42 | (52.5) | 38 | (47.5) | 80 |
| Origin | | | | | |
| Tipacoque | 17 | (26.6) | 47 | (73.4) | 64 |
| Soatá | 25 | (23.8) | 80 | (76.2) | 105 |
| Moniquira | 17 | (56.7) | 13 | (43.3) | 30 |
| Chitaraque | 22 | (38.6) | 35 | (61.4) | 57 |
| Miraflores | 8 | (72.3) | 3 | (27.7) | 11 |
| Otanche | 1 | (3.1) | 31 | (96.9) | 32 |
| San Pablo de Borbur | 0 | (0) | 6 | (100) | 6 |

**Table 2. Discordant results obtained from EC tests in patients from the department of Boyacá.**

| Participant Clinical Data | CDP | CSP | ELISA Chagas III GroupBios total antigens | ELISA IgG+IgM I Vircell recombinant antigens | IIF | Final Results according to chronic CD diagnostic algorithm [19] |
|---|---|---|---|---|---|---|
| Positive for *Leishmania* | Negative | Negative | Positive | Negative | NR | Negative |
| Positive for *Leishmania* | Negative | Negative | Positive | Negative | NR | Negative |
| Negative for *Leishmania* | Positive | Negative | Negative | Negative | N/A | Negative |
| Negative for *Leishmania* | Positive | Negative | Negative | Negative | N/A | Negative |

NR: Not reactive

N/A: Does not apply according to the diagnostic algorithm for CD

The sensitivity of the two RDT's were both 100%; while the specificity, PPV and DE obtained with the CDP tests were lower than those of the CSP tests (Table 4), due to the two false positive cases. The LR+ and LR- analyses showed that the RDT's can confirm and exclude diagnoses of CD. None of the RDT's presented cross reactions with samples of patients who were positive for leishmaniasis (Table 2).

The level of agreement of the CDP with the results of the reference tests was 99.5% and that of the CSP was 100% with excellent Kappa values (k = 99.1; CI95% 92.6–99.8%) and (k = 100; CI95% 94.3–100), respectively, demonstrating that the RDT's have a high degree of reproducibility.

## Discussion

During the chronic phase of CD, diagnosis is realized through the detection of IgG antibodies circulating in the bloodstream that are specific to *T. cruzi*. There are several serodiagnostic tests such as the ELISA, the IIF, and the indirect hemagglutination test (IHA), which are not available in the majority of the regions where the disease is endemic due to the complexities of carrying them out. Consequently, lack of a means of obtaining a diagnosis is one of the main obstacles for initiating treatment of CD [21]. Due to this, it is necessary to make new and simplified diagnostic tools such as RDT's available in order to decrease undiagnosed cases in areas in which diagnosis is not accessible through conventional techniques. In accordance with what has been written by other authors, RDT's must be validated in the field in every site in which they will be used in order for them to be recommended for the diagnosis of CD [39,25].

**Table 3. Two by two table for the results of the RDT and conventional serological tests for diagnosis of CD.**

| CDP | Reference test | | |
|---|---|---|---|
| | Positives | Negatives | Total |
| Positives | 90 | 2 | 90 |
| Negatives | 0 | 213 | 215 |
| Total | 90 | 215 | 305 |
| CSP | Reference test | | |
| | Positives | Negatives | Total |
| Positives | 90 | 0 | 90 |
| Negatives | 0 | 215 | 215 |
| Total | 90 | 215 | 305 |

**Table 4. Indices of validity and security for RDT's for the diagnosis of CD.**

| Indices | CDP | CSP |
|---|---|---|
| Sensitivity % (CI 95%) | 100 (95.9–100) | 100 (95.9–100) |
| Specificity % (CI 95%) | 99.1 (96.6–99.8) | 100 (98.3–100) |
| ED % (CI 95%) | 99.3 (97.1–100) | 100 (98.3–100) |
| PPV % (CI 95%) | 97.8 (92.3–99.7) | 100 (95.9–100) |
| NPV % (CI 95%) | 100 (98.2–100) | 100 (98.3–100) |
| LR+ | 107.4 | 9999.0 |
| LR- | 0.0 | 0.0 |

Diagnostically accurate studies have been reported with RDT's which conferred different results according to geographic location and the context in which they were carried out. In the present study, which was developed in endemic and non-endemic areas for *T. cruzi* in the department of Boyacá, the quality of the results obtained in terms of their validity as determined by their sensitivity and specificity compared to the results of reference tests were excellent for the two RDT's, the values for InBios CDP were 100% and 99.1%, respectively, and coincided with results reported in other studies using whole blood, such as a study done in Bolivia in 2014, which had a sensitivity of 96.2% and a specificity of 98.8% [25] and a study done in Chile with values of 99.3% for both measures, as well as the results of agreement with coventional serological tests, which was 98.2% (279/284), similar to the values found in the present study, which was 99.3% (303/305) [40].

As for the CSP, there was a sensitivity and specificity of 100%, which was similar to that described in Latin American migrants to Switzerland, who had values of 95.2% and 99.9% [41]; in Argentina with values of 95.3% and 99.5% [42]; in Bolivia, according to information from Rody et al. in 2008 with values of 93.4% and 99.0% [43] and by Lozano et al. in 2019 with values of 97.7% and 97.4%, respectively [34], they described DE of 97.5% similar to the results of the present study in that CSP was able to correctly classify 100% of the participants evaluated.

There have been reports of sensitivity and specificity of CDP and CSP that were lower than those obtained in the present study, such as in a multicentric study done on blood serum from 9 countries including Colombia, with values of 92.9% and 87.2%, respectively [44] and according to that reported by Reithinger et al., in which serum from four countries was analyzed, and average CDP sensitivity was 84.8% [45]. Also, for Verani et al., the sensitivity of CDP was 90.7% and that of CSP was 87.5% [39]. These differing results could have been influenced by the environment in which the RDT's were done (laboratory or field tests), for the nature of the samples utilized in the evaluation (due to how the serum was stored), or due to the antigenic variation of the strains of the parasite in the different regions in which the CD was transmitted [44].

Other parameters for determining the quality of the diagnostic tests in terms of safety are the PPV and the NPV, which provide estimates of probability of disease [46]. As it is important to interpret results in the context of the prevalence of the disease [47], these diagnostic indices have considerable inverse variation, as increasing prevalence increases PPV and decreases NPV and vice-versa [46]. In the present study, the results obtained in the PPV in CDP were 97.8%, and they were 100% for CSP; and in the NPV the two RDT's were 100%, observing that the two RDT's presented a high probability of returning a correct diagnosis, which coincides with the findings of Eguez et al. [26] and Lozano et al. [34]. The likelihood ratio results showed that the RDT's correctly measured the probability of a concrete result (positive or negative) according to the presence or absence of CD.

The etiological agents of the *T.cruz*i infection and of *Leishmania* spp. have a very close common phylogeny and share significant quantities of antigenic characteristics. Due to this, patients with one of the two infections, or with mixed infections may be misdiagnosed due to crossed serological reactions [48]. Thus, in the present study participants were included from the leishmaniasis endemic zone, as well as participants that were infected with leishmaniasis, of which, the RDT's did not present cross reactions, which concurred with results described by Lorca et al. [40] in that they did not find cross reactions with CDP and by Luquetti et al., who did not find cross reactions with CSP [24]. Among the limitations of this study, it should be mentioned that a reduced number of cases of leishmaniasis (38) were included, highlighting the importance of completing other studies in regions in which this parasite coexists with *T cruzi*.

In the reference tests, cross reactivity was found between the ELISA Chagas III GrupoBios test in two patients with leishmaniasis. This result was similar to that reported by another study in which a cross reaction was described that varied between 30 and 83% in conventional tests in which the antigen could be the entire parasite or soluble or purified extracts, the composition of which is a complex mix of antigens [47]. According to the diagnostic algorithm used in Colombia, when there is disagreement between the two ELISA tests, a third test is done. In the present study, the IIF titles were obtained with very close fluorescence, but were below the reference value [18]. According to the suggestions given, the best strategy for defining the significance of these discrepancies is long-term serological follow-up [47].

The Kappa index is a coefficient that is recommended as a measurement of the agreement of a test, adjusted for randomness [49]. Results similar to those found in the present study were described by Egüez, et al.; who reported a Kappa index of 0.99 for the two RDT's [26]; for CSP, Barfield reported 0.97% [42] and Shah et al. reported 0.94% for CDP in whole blood [25]. The results obtained in the present study showed that the RDT's have a high degree of reproducibility.

In several studies [26,34,40,50], the results of evaluation of the diagnostic exactness of the RDT's have suggested the use of these tests for the detection and monitoring of CD. These tests deliver results rapidly without the need for electrical equipment, utilizing small volumes of whole blood for samples, and can be completed anywhere (that is to say, in any region). They are highly recommended for primary care sectors in which laboratories are scarce or nonexistent, as well as for epidemiological monitoring programs or studies [44]. Although the RDT's can be stored at ambient temperatures between 8–30˚C during their useful lifespans, it is important to keep in mind that exposing them to temperatures greater than 30˚C may affect the performance of the tests. The cassette devices must be used immediately after their removal from the bag in order to minimize their exposure to humidity. [29,30]. Both RDT's evaluated in the present study complied with the ASSURED (Affordable, Sensitive, Specific, User-friendly, Rapid and robust, Equipment-free and Deliverable to end-users) criteria which are used as references for the identification of diagnostic tests that are most appropriate for areas with limited resources [51].

RDT's are recommended as diagnostic tools for chronic *T. cruzi* infection [34,50], however, current recommendations require the confirmation of positive results through conventional reference laboratory tests in which results may take a long time to be returned. Verifying the efficacy of two highly performing RDT's based on ensembles of distinct antigens such as CDP (multi-epitope antigens) and CSP (recombinant proteins) makes it possible to adapt them to be used as alternatives to conventional methodologies and diagnostic protocols for the epidemiological realities of each region. In the cases in which the RDT's return discordant results, as occured in the present study with 2 of the 305 cases tested, it is necessary to use a third test to confirm the result. Even though this increases costs, it should only constitute a small

percentage of the results, and is reasonable if the logistical costs of conventional serology are considered [26].

The findings of the present study show the excellent results that can be obtained using RDT's in terms of validity, safety, and reproducibility in the Colombian epidemiological context: they can be used in the field using samples of whole blood and they may be used in endemic areas in which laboratory installations necessary to realize conventional tests such as the ELISA, the IHA, or the IIF are unavailable; and in which infections with *T. cruzi* and leishmaniasis may be superimposed without presenting cross-reactions. The results obtained in the evaluation of precision diagnostics of the PDR's provide data that can be used for decision making in terms of the utilization of these tests within the diagnostic algorithm used in Colombia [18] for the diagnosis of chronic CD, thus improving access to treatment as quickly as possible beginning at the primary health care level.

## Supporting information

**S1 Verification list. Verification list of norms for the presentation of Diagnostic Precision Study Reports (STARD) in order to guarantee integrity and transparency.**
(DOCX)

**S1 Table. Complimentary table of qualitative and quantitative results of rapid diagnostic tests and standard reference tests utilized in the study.**
(XLSX)

## Acknowledgments

Thanks to all the participants included in the study, to the personnel of the Empresas Social del Estado of the seven municipalities included in the study, to the vector technicians of the CD monitoring program of the Secretaria de Salud de Boyacá for their help, and to Karina E. Egüez for the help she provided.

## Author Contributions

**Conceptualization:** Sandra Helena Suescún-Carrero.

**Data curation:** Sandra Helena Suescún-Carrero, Lyda Pilar Salamanca-Cardozo.

**Formal analysis:** Sandra Helena Suescún-Carrero, Lluis Armadans-Gil.

**Funding acquisition:** Sandra Helena Suescún-Carrero.

**Investigation:** Sandra Helena Suescún-Carrero, Lyda Pilar Salamanca-Cardozo.

**Methodology:** Sandra Helena Suescún-Carrero, Lluis Armadans-Gil.

**Project administration:** Sandra Helena Suescún-Carrero.

**Resources:** Sandra Helena Suescún-Carrero.

**Software:** Sandra Helena Suescún-Carrero, Lluis Armadans-Gil.

**Supervision:** Sandra Helena Suescún-Carrero, Lluis Armadans-Gil.

**Validation:** Sandra Helena Suescún-Carrero, Lyda Pilar Salamanca-Cardozo.

**Visualization:** Sandra Helena Suescún-Carrero, Lluis Armadans-Gil.

**Writing – original draft:** Sandra Helena Suescún-Carrero, María-Jesus Pinazo, Lluis Armadans-Gil.

**Writing – review & editing:** Sandra Helena Suescún-Carrero, María-Jesus Pinazo, Lluis Armadans-Gil.

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
