## [Decision Letter · Decision Letter 0]

11 Feb 2021

Dear dr Sandra Suescún-Carrero,

Thank you very much for submitting your manuscript "Evaluation of two rapid tests for the diagnosis of Trypanosoma cruzi infection in a Colombian population Evaluation of two rapid tests for Trypanosoma cruzi" for consideration at PLOS Neglected Tropical Diseases. As with all papers reviewed by the journal, your manuscript was reviewed by members of the editorial board and by several independent reviewers. In light of the reviews (below this email), we would like to invite the resubmission of a significantly-revised version that takes into account the reviewers' comments. 

We want to note in addition to reviewers' comments that the paper should be modified according to the STARD guidelines for the reporting of diagnostic studies (see https://www.equator-network.org/wp-content/uploads/2015/03/STARD-2015-checklist.pdf), beginning from the title and abstract. Be more coincise but linked to guidelines. 

Methodology seems solid but reporting should be improved and the issue of the leishmania positive samples should be explained well (It seems that you added leishmania positive semples to the analysis but in fact you found some individuals with positive serology (?) to leishmania in the study population).

We cannot make any decision about publication until we have seen the revised manuscript and your response to the reviewers' comments. Your revised manuscript is also likely to be sent to reviewers for further evaluation.

Sincerely,

Andrea Angheben

Associate Editor

Alain Debrabant

Deputy Editor

Reviewer's Responses to Questions

**Key Review Criteria Required for Acceptance?**

**Methods**

-Are the objectives of the study clearly articulated with a clear testable hypothesis stated?

-Is the study design appropriate to address the stated objectives?

-Is the population clearly described and appropriate for the hypothesis being tested?

-Is the sample size sufficient to ensure adequate power to address the hypothesis being tested?

-Were correct statistical analysis used to support conclusions?

-Are there concerns about ethical or regulatory requirements being met?

Reviewer #1: The sample size was sufficient to ensure the quality of the results and the statistical analysis were correct. 

The work was well executed and contain important information about the use of rapid diagnostic tests (RDT) to facilitate the diagnosis of Chagas Disease (CD) in regions where the laboratory infrastructure is poor. 

The authors declare that among the samples used, some were positive for leishmania, but I did not find the serological criteria or methodology used to define positivity for leishmania. It would be important to make it very clear how they concluded with leishmania positivity.

Reviewer #2: The study was performed in seven municipalities with high endemicity for CD and included 2 areas endemic for leishmaniasis. Study design appears to be adequate and compared the accuracy of 2 RDTs with the ELISA and IFI techniques performed in laboratories that are certified by the Bolivian MoH. Study development was according to the approved protocol, including reaching the programed sample size. 

Sample size determination needs in my view a clarification on the meaning of the sentence: “a sample of 305 people was required that included individuals without infections by T. cruzi, but with leishmaniasis”. 

Also it is not clear if Otanche and San Pablo Borbur are co-endemic for CD and Leish. 

The clinical form of leishmaniasis should be indicated as well as what diagnostic test was used.

Please also clarify the sentence: “in the present study, the ELISA Chagas III GrupoBios� test was used, and as a complementary test, according to the recommendations of INS, the recombinant antigen or synthetic peptide ELISA test was used with a specificity of � 98%, after first using a Chagas ELISA IgG+IgM I Vircell� test”. Which test is applied first? 

Statistical analysis of standard indicators of performance such as sensitivity, specificity, efficacy and safety , positive and negative predictive values, positive likelihood ratio and the kappa coefficient appears to obtain consistent results.

All ethical requirements appear to have been observed.

Reviewer #3: The methodology for establishing positive infection (lines 165-177) are problematic and difficult to follow (although they are similar to what is unfortunately used with some frequency). % senstitivity is given for the GrupoBios test – how was this number determined (using what sets of samples?). What are the sensitivities of the Vircell and the “recombinant antigens or synthetic peptide” tests (perhaps this refers to the same thing?)? Why is one of these referred to as “complementary”? Although hard to understand, it sounds like 3 tests were done and if there were “conflicting results” (what does conflicting results mean – all not positive or negative?) then an IFA was done. Why is this referred to as a “proof of principal” test? What were the antigens used in the IFA? Lines 216-219 refer to negative and positive and indicate that no “indeterminate results” were observed. What constitutes “conflicting samples”? It should be more clear how +/- was determined and what would be defined as “indeterminate”. Ideally, a supplemental table of all the test results for each sample, including quantitative results, should be supplied. 

How was Leishmania infection determined?

Although difficult to understand the pre-screening protocol, it sounds like all sera got tested initially with 2 or 3 tests (this is not clear) and if positive on one of these, then an IFA was done and if the IFA was positive, then the serum was considered positive – and if negative, was excluded from testing with the RDA. It seems, as in many evaluations, “positive” sera are marked as positive if they are positive on multiple other tests while those that test positive on only 1 test, are excluded. However, this does not resolve the discordant sera – and means that sera are being selected for DRT testing ONLY if they are strongly positive on multiple tests. Borderlines – even if truly positive are not further evaluated. This skews the results for the RDT. This is a common way of doing these types of evaluations but it gives overly optimistic data on the tests being evaluated – 100% sensitivity in this case. Which is quite unlikely given that the tests used to initially grade the sera are only 98% sensitive or less. At a minimum, the authors should note that the evaluated tests have 100 sensitivity using a highly selected set of sera.

**Results**

-Does the analysis presented match the analysis plan?

-Are the results clearly and completely presented?

-Are the figures (Tables, Images) of sufficient quality for clarity?

Reviewer #1: Results were clearly and completely presented, but I think that Legends of tables #1 and #2 can be improved to facilitate the interpretation.

Reviewer #2: Results are clearly presented. They show that the 2 RDTs performed well in the study sites, with high specificity, sensitivity, PPV, and NPV. When compared to the “gold standard” techniques, the RDTs showed high concordance and fewer discrepancies. Cross reactions in patients with leishmaniasis were not observed. 

The results of the present study favorably compare to other existing studies of RDTs’ performance in different endemic and non-endemic regions.

Reviewer #3: (No Response)

**Conclusions**

-Are the conclusions supported by the data presented?

-Are the limitations of analysis clearly described?

-Do the authors discuss how these data can be helpful to advance our understanding of the topic under study?

-Is public health relevance addressed?

Reviewer #1: It is important to reinforce the quality control conditions in which the work was carried out.

Reviewer #2: Conclusions are supported by the data. The limitations of the study are indicated, including the low number of patients with leishmaniasis. 

The data obtained contributes to the possible implementation of better performing diagnostic tests and algorithms for CD. 

The potential and benefit of introducing the studied RDTs in the Colombian setting are discussed, that could result in better patient management and reduced morbidity and mortality.

Reviewer #3: (No Response)

**Editorial and Data Presentation Modifications?**

Reviewer #1: I recommend "Minor Revision" before accept the article to publishing.

Reviewer #2: (No Response)

Reviewer #3: Line 6: needs rephrasing – 2 diagnostic tests are recommended, not all are ELISAs and not all have high sensitivity and specificity (the authors also make these points in lines 95-97

Line 76 “individuals”

**Summary and General Comments**

Reviewer #1: There are really different results in the literature regarding rapid diagnostic tests for the detection of anti-t.cruzi antibodies. As the authors themselves say, this can be the result of several factors. “These differing results could have been influenced by the environment in which the RDT's were done (e.g. laboratory or field tests), for the nature of the samples utilized in the evaluation (e.g. due to how the blood was stored), or due to the antigenic variation of the strains of the parasite in the different regions in which the CD was transmitted”

Furthermore, we consider that an important point must be taken into account: which quality control criteria were used as a reference in this study?. According to the mentioned logarithm, the reference was made to methodologies (two ELISAS and 01 IFI) but not quality control criteria was adopted. Some works cited actually used strict CC criteria. For example, in reference #44, a blind sample panel was used by all laboratories that participated in the evaluation. The authors cite that 4 different analysts performed the tests. Does this laboratory participate in external quality assessment programs for anti-t.cruzi tests? Does this laboratory have an internal quality control, with samples of low reactivity, of use to validate daily runs?

It would be interesting if the authors comment two aspects, regarding the positivity of the leishmania positive samples and the reinforcement of quality control procedures adopted. Perhaps could add some information about them in the text. Although the work should be published.

Reviewer #2: This is a relevant, correctly designed and implemented study that show consistent results. Further implementation research on the use of these RDTs is in my opinion warranted. 

The findings of this study are of interest for other NTDs. Optimization of CD diagnostics is also of interest for clinical practice in general both in endemic and non-endemic countries. 

The authors may want to consider some suggestions:

Transmission of CD via transplantation is missing. 

Use “patient management” instead of “patient treatment” (not all CD patients need anti-parasitic treatment). 

We know that RDTs do not need refrigeration but what about stability and shell life under field temperature and humidity conditions?

A word on the other diagnostic tests for CD such as Western blot, chemiluminescence and molecular techniques could be useful, since they may play a role if the RTDs become routine. 

The authors may want to discuss how the two RDTS performance fit into the ASSURED criteria (see https://www.who.int/bulletin/volumes/95/9/16-187468/en/).

I am not an English native speaker myself, therefore I would however recommend checking the validity of some terms with an English native speaker.

Reviewer #3: It is not really clear why the study was done as these tests, a noted in the Discussion, have been evaluated in a number of other studies and with widely varying (and lower than the current study) sensitivities. This is yet another result - to what end/better understanding? The authors offer reasons why results in these different tests could differ, but make no conclusions that would help in REALLY knowing the utility of these tests.

PLOS authors have the option to publish the peer review history of their article (what does this mean?). If published, this will include your full peer review and any attached files.

Reviewer #1: No

Reviewer #2: No

Reviewer #3: No
---

## [Decision Letter · Decision Letter 1]

17 May 2021

Dear dr Sandra Suescún-Carrero,

We are pleased to inform you that your manuscript 'Sensitivity and Specificity of two rapid tests for the diagnosis of infection by Trypanosoma cruzi in a Colombian population' has been provisionally accepted for publication in PLOS Neglected Tropical Diseases.

Before your manuscript can be formally accepted you will need to complete some minor changes on three points highlighted hereby and some formatting changes, which you can readwill receive in a follow up email. A member of our team will be in touch with a set of requests.

Best regards,

Andrea Angheben

Associate Editor

Alain Debrabant

Deputy Editor

From Associate Editor:

dear authors, many thanks for your patience and the work done to answer to reviewers remarks. I only ask you a little effort more, about three points:

- page 6 line 126: according to WHO recommendation and common practice for diagnosis of CD two tests based on differents techniques/antigens should be done. I ask you to add "/antigens" after "principles", I hope you agree

- page 7 line 158 after Methods title you inserted the doi of the protocol; I suggest to remove the link from this point adding a phrase after the title and before Study desing...concerning the protocol such as "The protocol of the study is available at the following link: doi..."

- finally page 8-9 in the section of tests methods please add a sentence concerning the blinded or unblinded way you used to perform the various tests: to me it would be more rigorous that you did the RDTs on field and the lab ELISAs would be done blind from the RDTs results. Was it like that? Please add a phrase on that.

Reviewer's Responses to Questions

**Summary and General Comments**

Reviewer #1: The article has been properly revised and deserves to be published.

The authors did a good job

---

## [Editor Report · Acceptance letter]

28 May 2021

Dear Mrs Suescún-Carrero,

We are delighted to inform you that your manuscript, "Sensitivity and Specificity of two rapid tests for the diagnosis of infection by Trypanosoma cruzi in a Colombian population," has been formally accepted for publication in PLOS Neglected Tropical Diseases.

Best regards,

Shaden Kamhawi

co-Editor-in-Chief

Paul Brindley

co-Editor-in-Chief
